**Data Availability Statement:** All relevant data are within the manuscript and its Supporting Information files.

# A live online exercise program for older adults improves depression and life-space mobility: A mixed-methods pilot randomized controlled trial

Giulia Coletta[1,2], Kenneth S. Noguchi[2,3], Kayla D. Beaudoin[1,2], Angelica McQuarrie[1,2], Ada Tang[2,3], Meridith Griffin[2,4], Rebecca Ganann[2,5], Stuart M. Phillips[1,2]*

1 Department of Kinesiology, McMaster University, Hamilton, ON, Canada, 2 McMaster Institute for Research on Aging, McMaster University, Hamilton, ON, Canada, 3 School of Rehabilitation Science, McMaster University, Hamilton, ON, Canada, 4 Department of Health Aging and Society and Gilbrea Centre for Studies in Aging, McMaster University, Hamilton, ON, Canada, 5 School of Nursing, McMaster University, Hamilton, ON, Canada

* phillis@mcmaster.ca

## Abstract

### Background

Aging is the primary risk factor for sarcopenia and mobility limitations. Exercise reduces these risks, but older adults have low levels of participation in physical activity and exercise. We investigated the preliminary effectiveness of a live, online exercise program on community-dwelling older adults' physical activity levels.

### Methods

A mixed-method pilot randomized controlled trial was conducted according to CONSORT 2010 statement: extension for pilot and feasibility trials. Sedentary older adults (65–80 years) were randomly assigned to the online exercise intervention (ACTIVE) or a waitlist control (CON) group. Outcomes were measured pre-randomization and following the 8-week intervention for ACTIVE and CON and two months following the end of the intervention for the ACTIVE group. Outcomes included habitual physical activity levels, depression, anxiety, loneliness, life-space mobility, nutrition risk, and feasibility. All participants were invited to participate post-intervention in individual semi-structured qualitative interviews. Reporting of the qualitative research followed the checklist for the Consolidated Criteria for Reporting research.

### Results

Seventeen older adults (71% women) were allocated to ACTIVE group, while 15 were allocated to CON group (87% women). Following the intervention, participants in the ACTIVE group reported reduced symptoms of depression (ACTIVE: pre = 4.2 ± 2.5; post = 2.2 ± 1.9; CON: pre = 3.5 ± 2.1, post = 3.5 ± 2.1; p <0.001) and improved life-space mobility (ACTIVE: pre = 62.4 ± 14.7; post = 71.8 ± 16.0; CON: pre = 65.1 ± 19.0, post = 63.6 ± 22.0; p = 0.003)

**Funding:** This work was supported by the Labarge Centre for Mobility in Aging within the McMaster Institute for Research on Aging Catalyst Grant (grant number: N/A, URL: https://mira.mcmaster.ca/research-centres/labarge-centre-for-mobility-in-aging/) to SMP. The funders had no role in study design, data collection and analysis, decision to publish, or preparation of the manuscript.

**Competing interests:** I have read the journal's policy and the authors of this manuscript have the following competing interests: SMP reports grants or research contracts from the US National Dairy Council, Canadian Institutes for Health Research, Dairy Farmers of Canada, Roquette Freres, Ontario Centre of Innovation, Nestle Health Sciences, Myos, National Science and Engineering Research Council and the US NIH during the conduct of the study; personal fees from Nestle Health Sciences, non-financial support from Enhanced Recovery, outside the submitted work. SMP has patents licensed to Exerkine but reports no financial gains from patents or related work. This does not alter our adherence to PLOS ONE policies on sharing data and materials

compared to waitlist control participants. The ACTIVE group had good adherence (97%) and acceptability (98%). Twenty-two participants participated in qualitative interviews. Five themes were identified, including (1) Feasibility of the online exercise program; (2) Perceived health benefits and improvements; (3) Registered Kinesiologists and Physiotherapists contributed to perceived safety; (4) Social connectivity associated with synchronous/live delivery; and (5) Growing old gracefully and preventing disability.

## Conclusions

Our online exercise program was acceptable to older adults, had good adherence, reduced depression, and increased life space. Participants reported improved functional and mental health benefits. Further research is warranted to expand on these findings.

## Trial registration

NCT04627493; 13/11/2020.

## Introduction

Age is a primary risk factor for developing chronic diseases and disability, including sarcopenia, frailty, and mobility limitations. Age-related diseases increase the risk of falls and hospitalization and diminish the quality of life for older adults [1,2]. Short periods of inactivity and increased time at home alone, imposed by surgery, illness, or public health mandates during viral pandemics, exacerbate the rate at which these age-related conditions progress [3,4]. Subsequently, there is an increased risk of developing secondary health conditions, including declines in mental health [5,6], physical health, and mobility [7–9], and increased risk of social isolation [3,5,7].

Physical activity (PA) plays an important role in offsetting or delaying age-related declines in functional capacity, quality of life, disability, and mortality [10–13]. Exercise and PA are effective in alleviating declines in strength and mobility [11,14,15], reducing anxiety [13] and depression [16], and preventing cognitive decline [10,11]. Critically, exercise-based intervention trials have been shown to reduce feelings of social isolation and loneliness [17,18], providing encouraging results, particularly given the strong association between lower levels of exercise and PA and the incidence of social isolation and loneliness [19]. However, older adults have described numerous barriers to participating in PA and exercise, including lack of access to places to exercise, feeling too anxious to be physically active, and requiring a large time commitment to travel to and from the exercise facility [20–22]. Implementing interventions utilizing online resources delivered by qualified exercise-trained professionals may be a viable service-delivery option versus traditional in-person exercise interventions [3]. Importantly, older adults do not prefer tools such as pre-recorded videos, which do not create a sense of social connectivity [7]. Thus, we proposed that employing a free access online meeting application (Zoom) to deliver live exercise classes led by qualified instructors and tailored to the needs of older persons would address physical inactivity in older persons [23].

The primary objective of this pilot randomized controlled trial (RCT) was to determine the preliminary effectiveness of a live online exercise program from qualified instructors on older adults' levels of PA compared to a waitlist control group. The secondary objectives were to examine the preliminary effectiveness of the online exercise program on older adults' mental

health, mobility, and nutrition and the program's feasibility, as well as determine if effects were maintained during the 2-month follow-up. We hypothesized that PA levels in individuals participating in the online exercise program would improve. We also hypothesized that the online exercise program may improve symptoms of mental health, life-space mobility, and nutrition from pre-to-post intervention and be maintained during the 2-month follow-up. We hypothesized that our live online exercise program would have good acceptability and adherence.

## Materials and methods

### Study design

This pilot convergent parallel mixed-methods RCT was guided by the CONSORT 2010 statement: extension for pilot and feasibility trials [24,25] (S1 File). The RCT compared community-dwelling older adults participating in a live online exercise program (ACTIVE) delivered on Zoom to a waitlist control group (CON). The intervention was carried out in three waves by Registered Kinesiologists and Physiotherapists from the Physical Activity Centre of Excellence at McMaster University in Ontario, Canada. Participants in the ACTIVE and CON groups underwent data collection (accelerometry and questionnaires) pre- and post-intervention. The three waves of the live online exercise intervention were delivered for the ACTIVE group between February 2021 and January 2022. Wave 1 classes occurred between February 2021 to April 2021 ($n = 8$), Wave 2 classes occurred between April 2021 to June 2021 ($n = 4$), and Wave 3 classes occurred between November 2021 and January 2022 ($n = 5$). During the 8 weeks that the ACTIVE group performed our online exercise intervention, the CON group were instructed to carry out their regular daily activity. Immediately following completion of the CON condition, participants in this group were offered the opportunity to partake in our live online exercise program for their respective waves (Wave 1, $n = 8$; Wave 2, $n = 3$; Wave 3, $n = 5$). The ACTIVE group completed follow-up data collection 2-months after completing the intervention. The 2-month follow-up was selected as evidence suggests that changes in steps can be seen ≤4 months following the completion of a study [26]. Feasibility measures (adherence and acceptability) were collected at each Zoom class using the poll feature. One month after completing the online exercise program, participants were invited to participate in the qualitative interviews. The duration of the study was from November 2020 to May 2022 and included the global COVID-19 pandemic (https://www.cihi.ca/en/canadian-covid-19-intervention-timeline), which had periods during which Public Health advised limiting close in-person interactions (stay home/lockdown) potentially affecting older persons mobility. The study was registered at clinicaltrials.gov (NCT04627493) and approved by the Hamilton Integrated Research Ethics Board (protocol #11429).

### Participants

Older community-dwelling women and men between the ages of 65 and 80 were recruited from Hamilton, Ontario, and the surrounding area. The upper age limit of 80 was chosen as mobility tends to decline far more quickly [27]; from a safety perspective, we wanted to limit the risks of our pilot program. Participants were recruited through newsletters, social media, and the local newspaper (Coffee News) between November 10th, 2020, and November 30th, 2021, which was the period during the global COVID-19 pandemic in which much of Hamilton, Ontario, was either completely closed or highly restricted. Potential participants were screened over videoconference (Zoom Video Communications Inc., 2016) to ensure they met the eligibility criteria, had access to the internet at home via a personal smartphone, tablet, or computer, and could safely participate based on the Canadian Society for Exercise Physiology

Get Active Questionnaire (https://csep.ca/2021/01/20/pre-screening-for-physical-activity/). Exclusion criteria included using a walker, cane, or assistive walking device; a history of neuromuscular conditions or muscle wasting diseases; and baseline participation in $\geq$150 minutes of moderate-to-vigorous PA per week. To be eligible for the qualitative portion of the study, participants needed to complete our online exercise program. Before providing a written informed consent form, all potential participants were informed of the study purpose, experimental procedures, and possible risks and provided the opportunity to ask questions.

## Intervention

Participants underwent an 8-week intervention consisting of thrice-weekly 60-minute online exercise classes (totalling 24 exercise classes) delivered online via Zoom. The blended group-based classes with 4–8 participants began with a 5-minute warm-up, 50 minutes of progressive strength, aerobic, and balance training focusing on functional movements, and a 5-minute cool-down. Strength exercises focused on lower (squats, hip abduction, hip extension, hamstring curls, front lunge, calf raises) and upper (biceps and triceps, shoulder press, front raise, side raise, alternating front and back fly) body exercises. Body weight was used for the first week, and household objects (e.g., water bottles, soup cans, pasta sauce) were introduced in subsequent weeks to increase the load. Participants were instructed to complete two sets of 8 repetitions, progressing to 12 repetitions per strength exercise. Aerobic exercises (marching, side steps, butt kicks, clock stepping, modified jumping jacks) were completed for 30 seconds for 2–3 sets. Participants progressed to 12 minutes of aerobic exercises by week 8. Balance exercises (side-by-side, semi-tandem, tandem, and single leg) were held for 15 seconds, and progressions included weight shifting, eyes closed, and dynamic balance. All exercises could be modified to be performed at two difficulty levels: basic and intermediate. The physiotherapist ($n$ = 1) and registered kinesiologists ($n$ = 3) trained to work with older adults supervised the classes and adjusted exercise intensity weekly to meet the participants' abilities. A research team member moderated each class to ensure participant safety and assist with any technological difficulties. All training sessions were completed in the participants' location of choice (e.g., participants' homes) and offered through the Physical Activity Centre of Excellence, a community gym for older adults at McMaster University.

## Outcome measures

All data were collected virtually. Demographic information, including gender, age, height, weight, race, highest level of education, marital status, living arrangement, and physical activity participation prior to the COVID pandemic, was collected at baseline. A scale and measuring tape were delivered to participants' homes to collect anthropometric measures (height, weight). Accelerometers were dropped off at participants' homes to wear for seven consecutive days and retrieved following this period. Questionnaires were administered online using LimeSurvey (LimeSurvey: An Open Source survey tool/LimeSurvey GmbH, Hamburg, Germany. URL http://www.limesurvey.org) pre-and-post-intervention for both the ACTIVE and CON and at a 2-month follow-up for the ACTIVE group. The individual qualitative post-intervention interviews were conducted via Zoom by the study lead (G.C.).

## Quantitative data collection

**Physical activity.** Habitual PA levels were the primary outcomes and were assessed using the armband BodyMedia Sensewear Pro II accelerometer (BodyMedia, Pittsburgh, PA, USA). Participants were instructed to wear the armband at all times, including while sleeping, and remove it only for brief periods for bathing or water activities. The armband was worn over

the left tricep muscle, and data were sampled in 60-second epochs from a heat flux sensor, a galvanic skin response sensor, a skin temperature sensor, a near-body temperature sensor, and a bi-axial accelerometer. Participant characteristics, including sex, age, height, weight, smoking status, and dominant hand, were used with accelerometer data to estimate daily steps and activity energy expenditure. The Sensewear armband accelerometer has been validated in free-living older adults [28]. The data was collected for seven consecutive days pre-and-post intervention for both the ACTIVE and CON and at a 2-month follow-up for the ACTIVE group.

**Mental health.** Depression, anxiety, and loneliness were included as secondary outcomes. The Geriatric Depression Scale Short Form (GDS-SF) assesses depressive symptoms using 15 questions regarding how participants felt over the past week (yes/no) [29]. Scores of >5 suggest depressive symptoms; >10 indicate moderate to severe depressive symptoms. The GDS-SF is a validated instrument for older adults [30]. The Geriatric Anxiety Inventory-Short Form (GAI-SF) measured anxiety symptoms using 5 items (e.g., "I often feel nervous" and "I worry a lot of the time") (yes/no). Scores between 0–8 reflect an absence of clinically significant anxiety; ≥9 reflects the presence of clinically significant anxiety [31]. The GAI-SF has good internal consistency and convergent and divergent validity for older (60–88 years) community-dwelling adults and is highly correlated with the original Geriatric Anxiety Inventory [32]. The 11-item Revised University of California Los Angeles loneliness scale (R-UCLA) measured loneliness using feeling isolated and available social connections [33]. The 11-item R-UCLA has good factorability and internal reliability in adults ≥65 years of age [33]. Scores range from 11 to 33, with higher scores indicating greater loneliness [33].

**Life-space mobility.** The University of Alabama Life Space Assessment, UA-LSA, was used to assess participants' life-space mobility during the month preceding the assessment [34] and has been validated in community-dwelling older adults [35,36]. The scores range from 0 (totally bed-bound) to 120 (travelled out of town daily without assistance).

**Nutrition risk.** The Seniors in the Community Risk Evaluation for Eating and Nutrition (SCREENII) is a valid and reliable tool to identify the risk for impaired nutritional states in community-living older adults [37]. The scoring of this tool ranges from 0 to 64; a score less than 50 is considered high nutrition risk.

**Feasibility.** Adherence was measured using the percentage of classes attended, and good adherence was defined a priori as >75% attendance (i.e., ≥19/24 sessions). Acceptability was measured using participant satisfaction using a 6-point Likert rating scale including: 1) very dissatisfied, 2) dissatisfied, 3) somewhat dissatisfied, 4) somewhat satisfied, 5) satisfied, and 6) very satisfied. Satisfaction was defined a priori as >60% of all ratings were satisfactory using a Zoom poll at the end of each call.

## Qualitative data collection

All participants (ACTIVE and waitlist CON) who completed our 8-week live online exercise program were invited via email to participate in an optional one-on-one 30-minute qualitative interview, regardless of group allocation. Semi-structured, open-ended 1:1 interviews were conducted and audio recorded via Zoom by the trained study lead (G.C., female) for a sub-sample of participants [38]. A qualitative interview guide (S1 Table) was developed with a qualitative methodology expert (M.G.). Questions were not provided or piloted with participants prior to the interview. The interviewer had a Bachelor of Science and was a Ph.D. student at the time of the interviews. There were no established relationships before the study commencement, and participants were aware that this research topic was of interest to the interviewer and part of the interviewer's Ph.D. dissertation. Participants were probed about their experiences and perceptions of exercise and our online exercise program [38]. Transcripts

were not returned to participants. Reporting of the qualitative research followed the checklist for the Consolidated Criteria for Reporting research (COREQ) [39].

## Sample size

A convenience sample size of 32 participants is aligned with previous online exercise pilot studies with sample sizes ranging from 10–44 [40,41]. For the qualitative interviews, we aimed to conduct between 20–24 interviews with a purposeful subsample of interested participants from the pilot RCT [42].

## Randomization, allocation concealment, and blinding

Eligible participants were randomized (block sizes from 4–8) to either the intervention group (ACTIVE) or the waitlist control group (CON). An investigator (S.M.P) generated the random allocation sequence using Research Randomizer (https://www.randomizer.org/). Participant group allocation was only provided to the study lead once baseline data collection was completed and immediately before the online exercise program started. Allocation concealment was attained in this study by ensuring a password-protected randomization schedule was accessed only by the investigator responsible for randomization and only at the time of randomization for each participant. G.C. was responsible for enrolling participants and advising them on group allocation. Given the nature of the study, it was impossible to blind the participants or investigators to the intervention.

## Analysis

**Quantitative.**  Descriptive statistics were calculated for demographic information and expressed as means ± standard deviation (SD) for continuous variables and counts (percent) for categorical variables. Pre-to-post-intervention changes in quantitative outcome measures were analyzed using a two-way repeated measures ANOVA once all data collection had been completed. The normality of the dependent variables and residuals was first assessed by visual inspection, followed by the Shapiro-Wilk test. If the assumptions of the ANOVAs were violated, data was log 10 transformed. A time comparison using a two-tailed paired t-test was used to analyze changes in quantitative outcome measures post-intervention to follow-up for the ACTIVE group. Missing data were deemed missing completely at random and addressed by multiple imputations [43]. The final intention-to-treat analysis included all randomized participants who met inclusion/exclusion criteria. A p-value <0.05 was considered statistically significant. Analyses were performed using IBM SPSS Statistics for Mac, version 28.0.

**Qualitative.**  Semi-structured interviews were thematically analyzed using the computer software Dedoose Version 9.0.17 (Los Angeles, CA, United States) [44]. The first two transcripts were coded independently by two coders (G.C. and K.S.N.), and initial patterns and themes were discussed to inform the preliminary coding book [24]. The coding book was shared and refined with input from a qualitative expert. The coders reviewed the subsequent transcripts using the coding book and iteratively revised them to reflect new concepts, meeting regularly to ensure consistency between interpretations. Themes and subthemes were inductively developed and refined with the research team. A third researcher was available if consensus could not be reached, but it was not required. Participants did not provide feedback on the findings.

## Results

### Participants' characteristics

Thirty-three older women and men participated in this study. Participant characteristics are presented in Table 1. There were no differences between groups for any variable. One participant in the ACTIVE group withdrew from the intervention due to an injury not attributed to the intervention. One participant in the CON group exceeded the 150 moderate-to-vigorous PA per week at baseline and, therefore, did not meet the eligibility criteria and was removed from the analysis. Missing data was deemed missing completely at random. Linear regression was used to impute missing data; thus, analyses for the ACTIVE and CON groups are $n = 17$ and $n = 15$, respectively (Fig 1).

### Habitual physical activity levels

There were no differences between groups for daily steps and daily active energy expenditure measured by the accelerometer following the intervention compared to the CON (Table 2). However, a main effect of time was observed from pre-to-post intervention for daily steps and active energy expenditure ($p < 0.05$). The data were not normally distributed for daily active

**Table 1. Descriptive characteristics of the study participants (N = 32).**

|  | ACTIVE n = 17 | CON n = 15 |
|---|---|---|
| **Women,** n (%) | 12 (70.6%) | 13 (86.7%) |
| **Age,** years | 70 ± 4 | 72 ± 5 |
| **Height,** cm | 165 ± 14 | 166 ± 7 |
| **Weight,** kg | 82 ± 17 | 80 ± 12 |
| **BMI,** kg/m$^2$ | 30 ± 5 | 29 ± 4 |
| **Race,** n (%) |  |  |
| Caucasian | 15 (88.2%) | 15 (100%) |
| Asian | 1 (5.9%) | 0 (0%) |
| Indigenous | 1 (5.9%) | 0 (0%) |
| **Highest level of education earned,** n (%) |  |  |
| Some school or high school diploma | 2 (11.8%) | 2 (13.3%) |
| Some college, vocational or training school | 1 (5.9%) | 1 (6.7%) |
| College graduate or bachelor's degree | 8 (47.1%) | 9 (60.0%) |
| Post-graduate training or degree | 6 (35.3%) | 3 (20.0%) |
| **Marital status,** n (%) |  |  |
| Never married | 0 (0.0%) | 2 (13.3%) |
| Divorced or separated | 4 (23.5%) | 2 (13.3%) |
| Widowed | 4 (23.5%) | 3 (20.0%) |
| Presently married | 9 (52.9%) | 8 (53.3%) |
| **Living arrangement,** n (%) |  |  |
| Wife, husband, or partner | 9 (52.9%) | 8 (53.3%) |
| Children | 1 (5.9%) | 0 (0.0%) |
| Friend(s) | 0 (0.0%) | 1 (6.7%) |
| Live alone | 6 (35.3%) | 6 (40.0%) |
| Other | 1 (5.9%) | 0 (0.0%) |
| **Participated in activity before the COVID pandemic,** n (%) | 9 (52.9%) | 9 (60.0%) |

*Notes*: Values are mean ± SD unless otherwise indicated. Abbreviations: CON = control; BMI = Body Mass Index.

### CONSORT 2010 Flow Diagram

**Fig 1. Consort flow diagram.** ITT = Intention to treat.

energy expenditure, and were transformed. The log 10 transformation did not change the interpretation; therefore, the original data is presented in Table 2. All ACTIVE and CON group participants complied with the full 7-day protocol pre-intervention, and 28 participants complied post-intervention. At the two-month follow-up, the ACTIVE group showed no significant differences from post-intervention ($p > 0.05$; Table 3). 14 participants complied with

**Table 2. Pre- and post-intervention daily average steps and active energy expenditure measured by accelerometers (N = 32).**

| Outcome | ACTIVE | | CON | | Time * Group | Time | $\eta_p^2$ |
|---|---|---|---|---|---|---|---|
| | Pre | Post | Pre | Post | | | |
| Daily Steps | 3599 ± 1506 | 4267 ± 2561 | 2895 ± 1749 | 3388 ± 1816 | 0.721 | 0.024 | 0.004 |
| Daily Active Energy Expenditure | 161± 178 | 230 ± 241 | 73 ± 94 | 132 ± 172 | 0.812 | 0.003 | 0.002 |
| Wear Time, % | 86.2 ± 0.2 | 85.4 ± 0.2 | 77.3 ± 0.2 | 82.1 ± 0.2 | - | - | - |

*Notes*: Values are mean ± SD.

the full 7-day protocol in the ACTIVE group and wore the accelerometer for 85.8 ± 0.2% of the time.

## Mental health

We observed an interaction (p = 0.012) for the Geriatric Depression Scale where the ACTIVE group decreased from pre-to-post intervention (Table 4). There were no significant between or within-group differences for symptoms of anxiety and loneliness (Table 4). The data were not normally distributed for GAI-SF, and were transformed. The log 10 transformation did not change the interpretation; therefore, the original data is presented in Table 4. Most participants in the ACTIVE and CON groups had scores less than 5 (absence of clinically significant anxiety) on the GAI-SF. At the two-month follow-up, the ACTIVE group showed no significant differences from post-intervention in depressive symptoms, anxiety symptoms, and loneliness (p > 0.05; Table 3).

## Life-space mobility

We observed an interaction (p = 0.014) for life-space mobility where the ACTIVE group increased from pre-to-post intervention (Table 4). At the two-month follow-up, the ACTIVE group maintained the difference from post-intervention on the life-space assessment (p > 0.05; Table 3).

## Nutrition risk

There were no between-group differences in nutrition risk (Table 4); scores lower than 50 indicated greater risk. However, a main effect of time was observed from pre-to-post intervention (p = 0.014). Pre-intervention, participants in both the ACTIVE (*n* = 14; 82%) and CON

**Table 3. ACTIVE group post to 2-month follow-up for primary and secondary outcomes (n = 17).**

| Outcome | Post | 2-month follow-up | p-value |
|---|---|---|---|
| Daily Steps | 4267 ± 2561 | 4171 ± 2134 | 0.679 |
| Daily Active Energy Expenditure | 230 ± 241 | 209 ± 243 | 0.476 |
| GDS-SF | 2.2 ± 1.9 | 2.0 ± 1.9 | 0.587 |
| GAI-SF | 1.4 ± 2.1 | 1.5 ± 2.2 | 0.908 |
| R-UCLA Loneliness | 16.2 ± 4.4 | 15.9 ± 3.7 | 0.539 |
| Life-Space Mobility | 71.8 ± 16 | 72.0 ± 15.9 | 0.332 |
| SCREENII | 47.6 ± 6.5 | 47.1 ± 6.9 | 0.616 |

*Notes*: Values are mean ± SD. Abbreviations: GDS-SF = Geriatric Depression Scale Short Form; GAI-SF = Geriatric Anxiety Inventory Short Form; SCREENII = Seniors in the community: Risk evaluation for eating and nutrition, Version II.

**Table 4. Pre- and post-intervention secondary outcomes (N = 32).**

| Outcome | ACTIVE | | CON | | Time * Group | Time | $\eta_p^2$ |
|---|---|---|---|---|---|---|---|
| | Pre | Post | Pre | Post | | | |
| **GDS-SF** | 4.2 ± 2.5 | 2.2 ± 1.9 | 3.5 ± 2.1 | 3.5 ± 2.1 | 0.012 | 0.012 | 0.193 |
| **GAI-SF** | 2.2 ± 2.7 | 1.4 ± 2.1 | 3.7 ± 3.8 | 3.0 ± 3.4 | 0.984 | 0.062 | 0.000 |
| **R-UCLA Loneliness** | 17.2 ± 3.9 | 16.3 ± 4.4 | 18.1 ± 4.9 | 17.3 ± 4.8 | 0.819 | 0.128 | 0.002 |
| **Life-Space Mobility** | 62.4 ± 14.7 | 71.8 ± 16.0 | 65.1 ± 19.0 | 63.6 ± 22.0 | 0.014 | 0.073 | 0.185 |
| **SCREENII** | 44.4 ± 7.1 | 47.6 ± 6.6 | 45.9 ± 7.1 | 47.1 ± 5.7 | 0.263 | 0.014 | 0.042 |

*Notes*: Values are mean ± SD. Abbreviations: GDS-SF = Geriatric Depression Scale Short Form; GAI-SF = Geriatric Anxiety Inventory Short Form; SCREENII = Seniors in the community: Risk evaluation for eating and nutrition, Version II. A p value of < 0.05 is significant.

($n$ = 12; 80%) groups had scores below 50 at baseline, indicating that our participants were at high nutrition risk at study entry. At the two-month follow-up, the ACTIVE group showed no significant differences in the SCREENII (p > 0.05; Table 3).

## Feasibility

Attendance rates were 97% in the ACTIVE group, ranging from 22/24 to 24/24 classes. The two most common reasons for non-attendance were illness ($n$ = 1 missed session in 4 participants) and health appointments ($n$ = 1 missed session in 3 participants). No adverse health events were reported throughout the study. Overall, participants reported being either satisfied or very satisfied with the intervention 98% of the time. Four participants reported being 'somewhat satisfied' between 1–4 times. Both adherence and acceptability were feasible based on the a priori definitions.

## Qualitative

A sub-sample of participants who completed our online exercise program ($N$ = 22; ACTIVE, $n$ = 12; and CON, $n$ = 10) participated in 30-minute interviews. Five overarching themes and eight sub-themes were identified related to perceptions, experiences, and feasibility of the program. These themes included the feasibility of the online exercise program, perceived health benefits and improvements, Registered Kinesiologists and Physiotherapists contributed to perceived safety, social connectivity associated with synchronous/live delivery, and growing old gracefully and preventing disability. Illustrative quotes for themes and sub-themes are presented in Table 5.

**Feasibility of the online exercise program.** Qualitative data reinforced our quantitative findings, conveying that the program had good acceptability and feasibility. Sub-themes were identified: 1) Acceptability of instructors, 2) Acceptability of exercises, and 3) Acceptability of time/location. Participants highlighted their satisfaction with the instructors and their age-appropriate exercise modifications.

**Perceived health benefits and improvements.** Participants described perceived health benefits and improvements after completing the online exercise program. Sub-themes included: 1) Improved function, 2) Improved chronic disease management, and 3) Improved energy and overall health. The qualitative data contradicted our quantitative findings; participants described walking more, feeling better physically and mentally, and perceived benefits in chronic disease management.

**Registered Kinesiologists and Physiotherapists contributed to the perceived safety of the intervention.** Participants perceived the program as safe, attributing it to the Registered

**Table 5. Qualitative, data-derived themes of participants' experiences and perceptions of exercise and the online exercise program (n = 22).**

| Theme | Subtheme & Quotes |
|---|---|
| **Feasibility of online exercise program** | **Acceptability of instructors** <br> ○ "I found the instructors, um just really excellent you know um respectful. I found them very respectful like, you know, there's sometimes people are not as respectful to older people and they don't encourage them for what they are doing well. . .so I thought just thought it was extremely I felt good and supported yeah I felt supported." Participant 16 <br> ○ "I thought the instructors did a really great job of explaining the various exercises and demonstrating how to do the movement correctly and safely. I thought that was really well done. I thought it was well done the way they offered options for the various movements, so if someone had like a physical limitation, they would show how to you know, do the movement sitting down rather than standing up, I thought they did a really good job with that. There was a very gradual warm up and cool down, I thought that was well done." Participant 9 <br> ○ "I think the ladies gave us enough opportunity for modifications of the exercise that this didn't suit or that didn't suit. So I think that was handled nicely. Because we're all at a different stage of wellness or age. So I think that was an important piece, that there was adaptability that they offered." Participant 10 <br> **Acceptability of exercises** <br> ○ "I like that it offered a mix of different things, it was balance, it was strength and some flexibility there was a variety to it, I thought that was really good." Participant 10 <br> ○ ". . . Because when I first signed up and I thought it was going to be like an hour of aerobics. . . and I thought oh my God I'll die, you know. And then I thought that it was going to be an hour of balance and an hour of aerobics and an hour of strength a week. Yeah, so it was a nice surprise when it turned out to be a mix up the three every hour." Participant 1 <br> **Acceptability of time/location** <br> ○ "I thought it was good variety three times a week was paramount in my eyes, and the fact that we got to do it at home." Participant 37 <br> ○ "I like the fact that it was a regular you know, three times a week at a set time so that was kind of motivating because I knew okay this you know I'm kind of planning my day and I know that it's going to start with the class." Participant 9 <br> ○ "So. . .the fact that I could do that on zoom is great I think it's the best thing ever." Participant 29 |
| **Perceived health benefits and improvements** | **Improved function** <br> ○ ". . .I knew my balance wasn't as good as it should be, but even that short time from when the classes ended till now and not doing it, I can see the difference again going bad. . . and the other thing that was huge for me that I, you know when started doing it, it was like. Oh, this is just so stupid that the biggest thing for me besides the balance was hand and ankle circles, because both my hands and my feet are not good, so doing that improved my walking so much. It was just amazing." Participant 28 <br> ○ "So, when I first started walking, I could maybe walk half an hour and that would be around my block I can now walk much further and much longer, with greater ease so that's been the go-to activity. I have added more activity, so I've started. . . changing my walking so it's more hiking related." Participant 13 <br> ○ ". . . now after exercising whatever it, it can be right, because I can't walk two kilometers or whatever, but I have increased my capabilities to walk without pain and I think it really made a difference." Participant 29 <br> ○ "I've noticed a difference almost right away, like after the after the first class the next day, I was I had muscle soreness, but it was good soreness. . . I have noticed that my balance has improved quite a bit." Participant 33 |

*(Continued)*

**Table 5.** (Continued)

| Theme | Subtheme & Quotes |
|---|---|
| | **Improved chronic disease management** |
| | ○ "Probably the best benefit for me because it did affect my diabetes for the better and my numbers for the better. I needed to get my numbers down; my blood pressure numbers doing your program were great. Using your program, which I spent a long time trying to get that consistency. . . when those numbers were dead on, I was like so happy." Participant 29 |
| | ○ "For me, the balance component, I thought was very valuable because I do have this thing with vertigo that I get, and my balance is not great, and I noticed as I'm aging that my balance seems to be deteriorating, so I thought those balance exercises were really great." Participant 9 |
| | ○ "I've had a stroke what five six years ago that was stress related, but certainly I was out of shape. And I've been working very hard to get my knee strengthened because I, you know with a delay in everything there's even less likelihood of getting a knee replacement . . .even though I may be slightly overweight or have a bad knee I can still find things through your kind of program, you gave a lot of examples of how to exercise sitting down or leaning on something so a lot of alternatives, so I think that the motivation was to figure out what I could do." Participant 31 |
| | ○ "I know that exercise I need physical exercise my body feels better. I've been in exercise programs for COPD. . .. Yeah, my respirologist was very interested when she heard about this, so I could hear in her voice that she was pleased right with this." Participant 16 |
| | ○ "I was a little hesitant because, as I said, I've got this heart issue that sometimes brings on a bit of heart pain and so on, and I never experienced that with the study, so I was good the development, I think, helped with that." Participant 34 |
| | **Improved energy and overall health** |
| | ○ "I felt better. . . I felt more energetic." Participant 7 |
| | ○ "Physical and mental, you know, I just felt better actually" Participant 16 |
| | ○ "I need a little more prompting and pushing to get me there because I know it's good for me, I know it's good for me physically, I know it's good for me mentally." Participant 3 |
| | ○ "Exercising because you should it's because it does make you feel better, and you know, there are other things that you get out of it. . ." Participant 2 |
| | ○ "Being able to get up and do something. Knowing that I'm going to feel better afterwards was a big motivation." Participant 11 |
| | ○ "But, on the whole, I look forward to it, and even there's moments where I didn't, but I would get up, turn the computer on, and then I feel better." Participant 29 |
| **Registered Kinesiologists and Physiotherapists contributed to perceived safety** | ○ "I felt safe in the way that. The kinesiologist, all of you that are you're very well trained, and so I felt like I was in extremely good hands as far as it goes, with this program. You know what you're doing, you know what the body [needs] and different than somebody that's just trained to teach exercises and don't have any understanding of the body as well." Participant 16 |
| | ○ "I thought the instructors did a really great job of explaining the various exercises and demonstrating and how to do the movement correctly and safely. I thought that was really well done. I thought it was well done the way they offered options for the various movements so if someone had like a physical limitation, they would show how to you know, do the movement sitting down rather than standing up, I thought they did a really good job. . . with that." Participant 9 |
| | ○ "I think the [instructors] gave us enough opportunity for modifications of the exercise that this didn't suit or that didn't suit. So, I think that was handled nicely. Because we're all at a different stage of wellness or age so I think that was an important piece, that there was adaptability that they offered." Participant 10 |

*(Continued)*

**Table 5.** (Continued)

| Theme | Subtheme & Quotes |
|---|---|
| **Social connectivity associated with synchronous/ live delivery** | **Group-based exercise program fosters connections**<br>○ "Maybe something like that would probably motivate me again that there's interaction, it's real people and it's done in real-time." Participant 37<br>○ "I enjoyed it, and I didn't feel alone, because you know someone else's behind that little video box." Participant 29<br>○ "...Especially through the pandemic, it was a godsend, you know I'm so grateful that I saw this newsletter, and so, for every reason... physical and mental you know I just felt better actually I felt better being part of the group too like a feeling like some kind of connection living alone it's difficult to make consistent connections that you're you know that you're going to see these people at certain times." Participant 16 |
| | **Socializing is not an important aspect when exercising**<br>○ "So, for me personally, I wasn't really interested in socializing, I mean that sounds so terrible, but I mean that wasn't why I was participating." Participant 9<br>○ "For me it wasn't about the social like it wasn't like social hour. So, for me that wasn't important so much." Participant 10<br>○ "... I would just take a look and see the other people who were there, I really didn't know any of them, but I really that was not a need for me, you know it's nice to be part of a group but for me I have a lot of other networks that I'm part of, so you know I really I really wanted to limit that, because if you spend a lot of time there then it's taking away from the exercise piece of it and the actual function of the original function of the program, so I think it is good to really manage it and it's tricky sometimes." Participant 2 |
| **Growing old gracefully and preventing disability** | ○ "I'm not going to say grow old gracefully, but yes, I want to grow old and not be stuck in a wheelchair or nursing home or something like that. That's a disaster in my mind." Participant 10<br>○ "I want to beat the crap; I'm trying to. I want to be able to keep moving." Participant 4<br>○ "One of the most significant things for me in all of this was to get through Covid with the less disability, you know...that's what I kept thinking is I managed to get through this and I'm actually I see it, more in my walking and my golfing and the gardening stuff that I'm actually stronger than I was and that's shocking, you know just I'm thrilled with that." Participant 2<br>○ "What does motivate me is as I'm aging, I want to get healthier, and yeah that would really be I want to maybe see my grandkids get married you know things like that so as I age, I get more motivation." Participant 37<br>○ "Since I've retired, I'm trying to I was exploring trying to find some kind of routine that would keep me keep me active and so that was and then losing weight and trying to get be healthy for longevity." Participant 13 |

Kinesiologists and Physiotherapists who delivered it. A few participants expressed the importance of personalized training to ensure proper form and reduce the possibility of injury. The qualitative findings reinforce program feasibility as the instructors contribute to the program's acceptability.

**Social connectivity associated with synchronous/live delivery.** Participants described social connectivity associated with synchronous/live delivery was identified as a theme with positive and negative aspects. The sub-themes include: 1) Group-based exercise program fosters connections, and 2) Socializing is not an important aspect when exercising. Most

participants found that group-based exercise fostered connection, whereas a few described socializing as "not an important aspect of exercise."

**Growing old gracefully and preventing disability.** Many participants described the desire to grow old gracefully - maintain mobility and health while participating in meaningful activities - and prevent disability. The potential loss of independence and fear of disability with aging motivates participation in the program. Community-dwelling older adults who participated were motivated before the start of the program and actively sought out the program.

## Discussion

We investigated the preliminary effectiveness and feasibility of an online exercise program for community-dwelling older adults. The program did not affect participants' habitual PA levels but reduced depressive symptoms, improved life-space mobility, and had excellent adherence and acceptability.

Habitual PA did not change pre- to post-intervention or at the 2-month follow-up for the ACTIVE group. Daily steps and daily active energy expenditure may not have changed due to the decreased level of PA in older adults during the quarantine period of COVID-19, which was in place while the study was implemented [45]. However, participants perceived improved function (e.g., walking, strength, gardening), chronic disease management (e.g., reduced blood pressure, vertigo), and improved energy and overall health. Participants stated they "felt better" and "had more energy".

The between-group changes in the GDS-SF showed the preliminary mental health benefits of the program. Participants in the ACTIVE group had reduced symptoms of depression compared to the CON group post-intervention, which were maintained at follow-up. Pre-intervention, 58% of our participants scored below five on the GDS-SF, suggesting no depressive symptoms. Strong evidence suggests the benefits of aerobic and resistance exercise on mental health, particularly depression [16]. Evidence also suggests the benefits of exercise in improving anxiety and loneliness [19,46]; however, there were no between-group differences in these outcomes. Most participants scored below five on the GAI-SF at pre-intervention, indicating the absence of clinically significant anxiety, which may have resulted in a floor effect. Similarly, most participants scored low on the R-UCLA loneliness scale pre-intervention. Interestingly, participants qualitatively described the importance of social connectivity and their desire to connect with other participants in the group. For example, two participants desired continued connection with the group following the end of the program (i.e., exchanging email or phone numbers). In contrast, a few participants preferred to refrain from engaging in socialization, as they did not view this as an important aspect of exercise and felt their social needs were already met.

Our results from a life space perspective are consistent with pre-post exercise intervention (stretching, muscle strength, balance, and walking) for older women [47], in which these authors reported improved life-space mobility scores post-intervention. Our data showed that the intervention improved life-space mobility in the ACTIVE group. The online exercise program may have emboldened participants to navigate spaces outside their homes, neighbourhoods, or towns. Higher scores on life-space mobility are associated with higher quality of life, improved social participation, reduced cognitive impairment, improved physical health, and decreased nutritional risk [48,49]. However, participants' SCREENII scores were categorized as high nutrition risk. This score is associated with poor health outcomes for older adults (e.g., increased risk of hospitalization and mortality) [50]. Future research should focus on multi-component interventions.

Our program was feasible, with 97% of classes attended and 98% of participants were either satisfied or very satisfied with each class. Online exercise programs are novel and emerging

modalities for engaging community-dwelling older adults in exercise. There is evidence from different patient populations using online exercise to improve important outcomes, including falls, balance, and health conditions. However, many programs focus on exergaming [51] or virtual gyms [52] and do not include live, real-time instructors to support individualized adaptations. Participants described enjoying the social connectivity associated with synchronous/live delivery. Overall, the live instructors, the routine of the group-based online program, and accountability facilitated engagement and high satisfaction, resulting in high program adherence. Online exercise programs are viable service-delivery options, particularly when in-person programming is unavailable due to physical distance, space limitations, and/or public health measures encouraging reduced mobility to slow viral spread. Future work should focus on synchronous online resources to engage older adults in exercise programs.

The qualitative interviews highlighted participants' desire to grow old gracefully and prevent disability. Some participants described the fear of disability from aging as a motivator to exercise. Others described their desire to continue to play with grandchildren, maintain their activities of daily living, and become healthier as motivators. These findings are consistent with previous studies investigating older adults' influences, motivations, and barriers to PA engagement [53,54].

There are several limitations with our study; however, primarily, we acknowledge the small sample size. The study was a pilot and, therefore, underpowered to determine if there was an interaction for our primary outcome. However, we noted that participants reported perceived health benefits and improved physical function post-intervention. Another limitation of the study includes a lack of blinding for our assessors and participants to group allocation. Our study only included English-speaking individuals with internet access at home with a tablet, computer, or phone. Future studies should attempt to include marginalized populations to improve generalizability with different language options, including subtitles.

## Conclusions

Our online exercise program reduced symptoms of depression, improved life-space mobility, and demonstrated good feasibility and perceived health benefits. We did not see an effect of the intervention in the ACTIVE group on accelerometry-derived outcomes. Nonetheless, following participation in the online exercise program, participants perceived physical and mental health benefits. Given the feasibility of the methods and perceived benefits of the intervention, our online exercise program has the potential to support synchronous exercise programs at home. The findings will inform a larger RCT utilizing synchronous online exercise programming delivered by healthcare professionals with modifications to the protocol regarding progressions, longer training periods, and a larger sample size. Future work will focus on the effectiveness of group-based synchronous exercise programs on functional assessments.

## Supporting information

**S1 Table. Semi-structured interview questions.**
(DOCX)

**S1 File. CONSORT pilot or feasibility trial checklist.**
(PDF)

**S2 File. Minimal dataset.**
(XLSX)

**S3 File. Protocol.**
(PDF)

## Acknowledgments

We want to acknowledge and thank the Physical Activity Centre of Excellence (PACE) Registered Kinesiologists and Physiotherapists for their assistance in designing and delivering our online exercise program.

## Author Contributions

**Conceptualization:** Giulia Coletta, Angelica McQuarrie, Ada Tang, Meridith Griffin, Stuart M. Phillips.

**Data curation:** Giulia Coletta, Kayla D. Beaudoin, Stuart M. Phillips.

**Formal analysis:** Giulia Coletta, Kenneth S. Noguchi, Kayla D. Beaudoin, Rebecca Ganann, Stuart M. Phillips.

**Funding acquisition:** Giulia Coletta, Kenneth S. Noguchi, Ada Tang, Meridith Griffin, Stuart M. Phillips.

**Investigation:** Giulia Coletta, Stuart M. Phillips.

**Methodology:** Giulia Coletta, Stuart M. Phillips.

**Resources:** Angelica McQuarrie.

**Supervision:** Stuart M. Phillips.

**Writing – original draft:** Giulia Coletta.

**Writing – review & editing:** Giulia Coletta, Kenneth S. Noguchi, Kayla D. Beaudoin, Angelica McQuarrie, Ada Tang, Meridith Griffin, Rebecca Ganann, Stuart M. Phillips.

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
