## [Decision Letter · Decision Letter 0]

13 Sep 2024

PONE-D-24-24624A live online exercise program for older adults improves depression and life-space mobility: A mixed-methods pilot randomized controlled trialPLOS ONE

Dear Dr. Phillips,

Thank you for submitting your manuscript to PLOS ONE. After careful consideration, we feel that it has merit but does not fully meet PLOS ONE’s publication criteria as it currently stands. Therefore, we invite you to submit a revised version of the manuscript that addresses the points raised during the review process.

As noted by the Reviewers, the manuscript would benefit from the addition of more detail on the methods including the exercise intervention. As this is an RCT, it was reviewed by a statistician who has suggestions for revising the presentation of results. In addition to the comments of the 3 reviewers, I note the following: 1. It would help if a rationale was provided for the 2-month follow-up (i.e., how was this duration chosen?) and if a study objective was included related to this measurement period. Did the authors expect that participants would maintain benefits during the 2 month follow-up? If so, please include this hypothesis. 2. Line 115: "Feasibility measures" is mentioned here, but only one measure is listed in the brackets. 3. Line 125: Please clarify how you determined that participants were "healthy", and why the upper age limit of 80 was chosen. 4. Line 141: Please clarify if all 17 participants participated in the same online class. And over what months did the exercise class take place? I assume it is between Dec 2021 and May 2022, but it would help to clarify the exact timing within the overall study duration. Was the program offered through the university or a local community centre? How many physios/kinesiologists delivered the program? Were they asked any questions about how feasible they thought it was to deliver the program? 5. Please include the demographic outcomes in the methods. 6. Please clarify the PA outcomes in the methods, and provide more details on accelerometer data processing. Did all participants comply fully with the 7-day protocol? Did they record non-wear time? 7. Line 184: The subheading is Life-space mobility in the Results. 8. Line 190: I don't follow the rationale for including nutrition risk as an outcome. Was there a nutrition education component of the intervention that would impact this score? 9. Line 205: Do you know if any of the CON participants started an exercise program during the study period? 10. Line 235: Change to "Descriptive statistics were"11. Please provide more detail on the missing data and multiple imputation. Please clarify if the Figure 1 flowchart in reference 41 was followed prior to multiple imputation. Also, the flow diagram doesn't include the 2-month follow-up. Did all 16 ACTIVE participants return for the 2-month follow-up? 12. Line 263: I don't follow why the one CON participant wasn't excluded prior to randomization since they didn't meet the inclusion criteria. 13. In the figures, I would suggest providing the exact p-values above the plots instead of the "Dissimilar letters"14. Lines 295-296: Based on the values provided here, it seems that both groups demonstrated a decrease in symptoms of anxiety and loneliness - is that correct? If so, please clarify that in the text. Similarly, in Line 301, please clarify that this comparison is relative to the post-intervention timepoint.15. Line 340: How did you know that these 4 participants felt the exercises were too easy? Was there a place in the Zoom poll to enter reasons why they weren't fully satisfied? 16. Line 344: sorry for my confusion, but I thought only 16 participants completed the intervention. How then were there 22 interview participants? Does this include some of the CON participants after they received the intervention? As per the other reviewer's comment, please clarify when the CON participants received the intervention, and if all of them joined one exercise class. 17. Since this was a pilot study, it would be helpful to know how the authors plan to use these findings to inform a future trial, and if the intervention is continuing to be delivered or if there are plans to deliver it more broadly. It would also be helpful to know what the 18. References 6, 24 and 26 are incomplete. 

We look forward to receiving your revised manuscript.

Kind regards,

Heather Macdonald, Ph.D

Academic Editor

PLOS ONE

“I have read the journal's policy and the authors of this manuscript have the following competing interests: SMP reports grants or research contracts from the US National Dairy Council, Canadian Institutes for Health Research, Dairy Farmers of Canada, Roquette Freres, Ontario Centre of Innovation, Nestle Health Sciences, Myos, National Science and Engineering Research Council and the US NIH during the conduct of the study; personal fees from Nestle Health Sciences, non-financial support from Enhanced Recovery, outside the submitted work. SMP has patents licensed to Exerkine but reports no financial gains from patents or related work.”

4. We note that there is identifying data in the Supporting Information file <S2 File>. Due to the inclusion of these potentially identifying data, we have removed this file from your file inventory. Prior to sharing human research participant data, authors should consult with an ethics committee to ensure data are shared in accordance with participant consent and all applicable local laws.

-Location data

Reviewers' comments:

Reviewer's Responses to Questions

**Comments to the Author**

1. Is the manuscript technically sound, and do the data support the conclusions?

Reviewer #1: Partly

Reviewer #2: Yes

Reviewer #3: Yes

2. Has the statistical analysis been performed appropriately and rigorously? 

Reviewer #1: No

Reviewer #2: Yes

Reviewer #3: I Don't Know

3. Have the authors made all data underlying the findings in their manuscript fully available?

Reviewer #1: Yes

Reviewer #2: Yes

Reviewer #3: Yes

4. Is the manuscript presented in an intelligible fashion and written in standard English?

Reviewer #1: Yes

Reviewer #2: Yes

Reviewer #3: Yes

5. Review Comments to the Author

Reviewer #1: A brief description of the waitlist control is to be provided.

Line 173: Information on the validity of the tool GAI – SF is to be provided.

Line 219: While formal statistical power calculations are often not required for pilot studies, having a calculated sample size can enhance the study's ability to provide useful estimates to effectively inform the design for the main trial or when involved in some statistical analysis.

Line 231: Description of allocation concealment prior to allocation to the groups is to be provided.

Line 241: One-tailed or two-tailed test p-value is to be stated.

Line 241-242: The sentence requires revision.

Table 1: 1 decimal to be provided for the percentages.

Line 271-276, 293-301, 319-322, 326-327: Results are to be presented in table form.

Some data looks skewed. A statement on whether the data are normally distributed and the fulfillment of parametric test(s) assumptions are to be provided.

Line 319, 328: The main table output (test of within/between subjects) from the two-way repeated measure ANOVA analysis is to be included.

Line 350: From the qualitative study, the number of participants who took part in the discussion from the list of n=22 is to be mentioned.

In one-to-one interviews in qualitative studies, if no new themes emerge and existing themes are exhausted, will the interview stop even if the sample size has been met?

Reviewer #2: Robust PhD study and very well written.

I have suggestions that might help with understanding, replication, or translation into practice.

I would like to see more about what the intervention actually was. Especially since the intervention was described by authors as 'progressive' and participants mentioned the intervention 'offered options' but some participants also rated the exercises 'too easy.'

Was any equipment/household items used/needed for programme. Could you describe the progressive technique used? How could this programme be improved (if at all) for a full RCT? What could we learn from this study (from a practitioners view)?

Were all 17 participants in the same zoom class? Again, was this feasible, or recommended for future online classes? Did it hold some more advanced participants back (hence the 'too easy' rating)?

Participants are described as healthy. Do you have data on how many health conditions the participants had on average?

Reviewer #3: General feedback:

This is a mixed-methods pilot randomized controlled trial of N=32 participants, comparing the preliminary effectiveness of a live, online exercise program for older adults compared to a waitlist control group. The primary outcome was physical activity levels as measured by accelerometery. Overall, this is a well-designed study with clear and compelling writing. The authors demonstrated that this was a feasible program with high degree of program attendance and participant satisfaction. Although underpowered to detect differences the authors report improvements in depression and life-space mobility and bolster these findings with qualitative support.

Some limitations of the study include the use of a waitlist control vs. an attention control. Something is often better than nothing in exercise trials and it is not possible to understand if contextual factors are driving improvements vs. the exercise program itself.

Further, the online exercise intervention is not well reported. The authors describe 3x60 min of online exercise classes delivered via Zoom, 5 min warmup/cool down, 50 min of progressive strength, aerobic, and balance training focusing on functional movements. There are several important elements missing here such as: the actual exercises performed in each of the categories, exercise dosage (i.e. reps/sets/ RPE based?), decision rules for how exercises were progressed, use of exercise equipment etc. I would recommend referring to the CERT guidelines (consensus on exercise reporting template): https://bjsm.bmj.com/content/50/23/1428

Another limitation is that the primary outcome measure of “habitual physical activity” is not clearly defined in the methods- i.e. it is discussed that it will be measured using accelerometery, but it is not clear what exact metric you are considering your primary outcome i.e. daily steps vs. daily active energy expenditure.

Lastly, the qualitative analysis lacks sufficient detail in reporting which decreases confidence in the rigour of this aspect of the study.

Strengths:

-well-designed study

-compelling and clear writing

Limitations:

-online exercise is not well reported

-main outcome measure of “habitual physical activity” is not defined in methods.

-some challenges with qualitative analysis reporting

-blinding not possible (acknowledged in limitations)

-small sample size (acknowledged in limitations)

Line by line feedback:

Line 96: “ability-modified, low-risk exercise program”- this is the only time the exercise program is described in this way. What makes an exercise program high vs. low risk? Typically exercise interventions delivered by physiotherapists are described as “individually tailored” and parameters for how exercises were regressed or progressed are described.

Line 205: “All participants (ACTIVE and CON) who completed the online exercise program were invited via email to participate in an optional one-on-one 30-minute qualitative interview, regardless of group allocation.” I think it would be helpful to mention a bit more about the waitlist control group- when were these individuals invited to take part in the intervention? I don’t think it is mentioned prior to this statement.

Line 207: Indicate whether these were 1:1 interviews. Also did every participant in the trial participate in an interview? If not, perhaps state a “sub-sample of participants”.

Line 247: Your description of the qualitative analysis should include a reference for thematic analysis. Also, the analysis process is not clear… It appears that the first step was creating an initial codebook based on pre-specified topics of interest (deductive approach)- but the next sentence states that the coders met to discuss new codes identified and code the remaining transcripts. Were the first few interviews coded together, where the pre-specified topics and the data itself was used to create the initial codebook? If so, this should be specified and should clearly indicate the combination of deductive and inductive approaches. How did the coders ensure consistency between their interpretation i.e. intercoder reliability?

Line 252-256: This sentence is a bit confusing- by introducing the sentence with “this iterative process”, I assume you are talking about coding, but you are introducing the creation of themes and sub-themes. Revise to increase clarity and include a description of how the codes were crystalized into their preliminary and final themes.

Line 255: - “the themes and sub-themes were those most commonly cited in the data”- did the authors use actual code counts to derive themes and sub-themes?

Another point to address in your description of the qualitative analysis is how the interview guides were created and whether they were they piloted with participants?

Line 360: “The qualitative data refute our quantitative findings”… I think this statement is a bit too strong as the quantitative data lacks sufficient power to detect an effect with any certainty.

Line 379: I think you need to define what “growing old gracefully” means here.

6. PLOS authors have the option to publish the peer review history of their article (what does this mean?). If published, this will include your full peer review and any attached files.

Reviewer #1: No

Reviewer #2: No

Reviewer #3: No

---

## [Author Response · Author response to Decision Letter 0]

3 Oct 2024

We thank the editor and reviewers for their diligence and constructive comments on our work. We provide replies here, and a substantially revised manuscript that we hope addresses the points raised by the editor and reviewers and is now ready for publication. 

Editor: 

1. It would help if a rationale was provided for the 2-month follow-up (i.e., how was this duration chosen?) and if a study objective was included related to this measurement period. Did the authors expect that participants would maintain benefits during the 2 month follow-up? If so, please include this hypothesis. 

Thanks for raising this. We have included an updated rationale for the 2-month follow-up period and included the study objective and hypothesis. 

2. Line 115: "Feasibility measures" is mentioned here, but only one measure is listed in the brackets. 

We have included both adherence and acceptability in the bracket. 

3. Line 125: Please clarify how you determined that participants were "healthy" and why the upper age limit of 80 was chosen. 

We have removed the term healthy as we did not exclude medications or health conditions. We have included a rationale for choosing the upper age limit of 80. 

4. Line 141: Please clarify if all 17 participants participated in the same online class. And over what months did the exercise class take place? I assume it is between Dec 2021 and May 2022, but it would help to clarify the exact timing within the overall study duration. Was the program offered through the university or a local community centre? How many physios/kinesiologists delivered the program? Were they asked any questions about how feasible they thought it was to deliver the program? 

Thank you for these questions. We have clarified the months that the intervention took place, including the number of participants randomized to the ACTIVE group and participating in each wave of the study. We specified that the program was offered through a community gym for older adults at McMaster University. We also included the number of physios/kinesiologists who delivered the program. Unfortunately, we did not ask the physios/kinesiologists about the feasibility of delivering the program. 

5. Please include the demographic outcomes in the methods. 

We have included the demographic outcomes in the methods section under Outcome Measures. 

6. Please clarify the PA outcomes in the methods and provide more details on accelerometer data processing. Did all participants comply fully with the 7-day protocol? Did they record non-wear time? 

- equation for active energy expenditure 

We agree and have included more details on the PA outcomes and accelerometer data processing. We have included the number of participants who complied fully with the 7-day protocol and the wear-time in the description and Table 2 under the subheading Habitual physical activity levels. 

7. Line 184: The subheading is Life-space mobility in the Results. 

We have modified the subheading in the methods section to “Life-space mobility.”

8. Line 190: I don't follow the rationale for including nutrition risk as an outcome. Was there a nutrition education component of the intervention that would impact this score? 

We included the nutrition risk questionnaire as a screening measure to assess why some individuals may not have responded to the exercise intervention and also as an exploratory outcome. There was no nutrition education component of the intervention that would impact this score. 

9. Line 205: Do you know if any of the CON participants started an exercise program during the study period? 

We do not believe any CON participant began an exercise program during the study. Following the end of the study data collection period, the CON participants had the opportunity to begin our online exercise program. Therefore, they were invited to participate in the qualitative interviews after completing our 8-week live online exercise program. We have amended this sentence to communicate this better. 

10. Line 235: Change to "Descriptive statistics were"

Thank you. We have addressed this comment in the manuscript. 

11. Please provide more detail on the missing data and multiple imputation. Please clarify if the Figure 1 flowchart in reference 41 was followed prior to multiple imputation. Also, the flow diagram doesn't include the 2-month follow-up. Did all 16 ACTIVE participants return for the 2-month follow-up? 

In the analysis and results sections, we have provided more detail on the missing data and multiple imputation. The Figure 1 flowchart in reference 41 was used to help determine if multiple imputation should be used to handle missing data. We have updated the CONSORT flow diagram (Figure 1) to include more detail on the missing data, the 2-month follow-up, and the number of ACTIVE (n = 17) participants who returned for the 2-month follow-up. 

12. Line 263: I don't follow why the one CON participant wasn't excluded prior to randomization since they didn't meet the inclusion criteria. 

Data was only analyzed after the study was completed; therefore, we were unaware that the CON participant didn’t meet the inclusion criteria prior to randomization. We have included a sentence in the Quantitative Analysis section and Participants’ characteristics to clarify this point.

13. In the figures, I would suggest providing the exact p-values above the plots instead of the "Dissimilar letters"

As per reviewer 1’s comments, we have changed all figures to tables. 

14. Lines 295-296: Based on the values provided here, it seems that both groups demonstrated a decrease in symptoms of anxiety and loneliness - is that correct? If so, please clarify that in the text. Similarly, in Line 301, please clarify that this comparison is relative to the post-intervention timepoint.

We have clarified these points in the results section text. 

15. Line 340: How did you know that these 4 participants felt the exercises were too easy? Was there a place in the Zoom poll to enter reasons why they weren't fully satisfied?

We have removed this statement as it was not possible for participants to input this information in the Zoom poll. 

16. Line 344: sorry for my confusion, but I thought only 16 participants completed the intervention. How then were there 22 interview participants? Does this include some of the CON participants after they received the intervention? As per the other reviewer's comment, please clarify when the CON participants received the intervention, and if all of them joined one exercise class. 

We have included the number of participants from the ACTIVE and CON groups that participated in the qualitative interviews. We have also included a sentence in the Participants subheading to indicate that participants who completed our online exercise program were invited to participate in the qualitative interviews. Further details have been included under the Intervention heading, detailing when the waitlist control group participated in the intervention and the number of participants in each wave. 

17. Since this was a pilot study, it would be helpful to know how the authors plan to use these findings to inform a future trial, and if the intervention is continuing to be delivered or if there are plans to deliver it more broadly. It would also be helpful to know what the 

We have included information on how we plan to use these findings to inform a future trial. 

18. References 6, 24 and 26 are incomplete. 

We have updated the references. 

Reviewer #1: A brief description of the waitlist control is to be provided.

We thank the reviewer for their comments. We have included a brief description of the waitlist control in the Study Design subheading and other details throughout the manuscript. 

Line 173: Information on the validity of the tool GAI – SF is to be provided.

We included the validity of the GAI-SF tool in this population. 

Line 219: While formal statistical power calculations are often not required for pilot studies, having a calculated sample size can enhance the study's ability to provide useful estimates to effectively inform the design for the main trial or when involved in some statistical analysis.

Thank you for raising this point. We have not reported a post hoc sample size (to detect a significant effect), as many contest the validity of doing so. Effect sizes have been calculated and reported for each of the outcomes.

Line 231: Description of allocation concealment prior to allocation to the groups is to be provided.

A description of allocation concealment has been included under the heading Randomization, Allocation Concealment, and Blinding. 

Line 241: One-tailed or two-tailed test p-value is to be stated.

We included a statement in the analysis section indicating that a two-tailed test was used. 

Line 241-242: The sentence requires revision.

We have revised this sentence for clarity. 

Table 1: 1 decimal to be provided for the percentages.

We have included 1 decimal for all the percentages in Table 1. 

Line 271-276, 293-301, 319-322, 326-327: Results are to be presented in table form.

We have presented all data in Tables 2-4. 

Some data looks skewed. A statement on whether the data are normally distributed and the fulfillment of parametric test(s) assumptions are to be provided.

We have included a statement on the tests carried out to assess normality and the use of non-parametric tests.

Line 319, 328: The main table output (test of within/between subjects) from the two-way repeated measure ANOVA analysis is to be included.

Thank you; we have provided the main table output from the two-way repeated measure ANOVA. 

Line 350: From the qualitative study, the number of participants who took part in the discussion from the list of n=22 is to be mentioned.

We agree and have included the number of participants from the quantitative study who participated in the qualitative study and their respective groups under the Qualitative results subheading. 

In one-to-one interviews in qualitative studies, if no new themes emerge and existing themes are exhausted, will the interview stop even if the sample size has been met?

We’re unsure if we understand the reviewer’s question. We conducted and completed all the interviews before commencing data analysis. Therefore, we aimed to achieve a sample size between 20 and 24 people, as data saturation generally occurs within this range. 

Reviewer #2: Robust PhD study and very well written.

We thank the reviewer for their comments. 

I have suggestions that might help with understanding, replication, or translation into practice.

I would like to see more about what the intervention actually was. Especially since the intervention was described by authors as 'progressive' and participants mentioned the intervention 'offered options' but some participants also rated the exercises 'too easy.'

We have included more details in the intervention as per the CERT guidelines. 

Was any equipment/household items used/needed for programme. Could you describe the progressive technique used? How could this programme be improved (if at all) for a full RCT? What could we learn from this study (from a practitioners view)?

We have included further information on the equipment and progressive technique of the intervention under the Intervention subheading. Unfortunately, we do not have data on the practitioner’s view from this study. 

Were all 17 participants in the same zoom class? Again, was this feasible, or recommended for future online classes? Did it hold some more advanced participants back (hence the 'too easy' rating)?

Thanks for these questions. Under the Study Design subheading, we provided more details on how the Zoom classes were delivered. Briefly, three waves of classes were held for the ACTIVE group. Unfortunately, although this would have been interesting, we did not collect data from the Physiotherapists/Kinesiologists to understand the feasibility of facilitating classes on Zoom or the impact of group-based classes on progressions for individual participants. 

Participants are described as healthy. Do you have data on how many health conditions the participants had on average?

Unfortunately, we do not have information on the number of health conditions the participants had. We have removed the term healthy as per the suggestion of the editor. 

Reviewer #3: General feedback:

This is a mixed-methods pilot randomized controlled trial of N=32 participants, comparing the preliminary effectiveness of a live, online exercise program for older adults compared to a waitlist control group. The primary outcome was physical activity levels as measured by accelerometery. Overall, this is a well-designed study with clear and compelling writing. The authors demonstrated that this was a feasible program with high degree of program attendance and participant satisfaction. Although underpowered to detect differences the authors report improvements in depression and life-space mobility and bolster these findings with qualitative support.

Some limitations of the study include the use of a waitlist control vs. an attention control. Something is often better than nothing in exercise trials and it is not possible to understand if contextual factors are driving improvements vs. the exercise program itself.

Further, the online exercise intervention is not well reported. The authors describe 3x60 min of online exercise classes delivered via Zoom, 5 min warmup/cool down, 50 min of progressive strength, aerobic, and balance training focusing on functional movements. There are several important elements missing here such as: the actual exercises performed in each of the categories, exercise dosage (i.e. reps/sets/ RPE based?), decision rules for how exercises were progressed, use of exercise equipment etc. I would recommend referring to the CERT guidelines (consensus on exercise reporting template): https://bjsm.bmj.com/content/50/23/1428

Another limitation is that the primary outcome measure of “habitual physical activity” is not clearly defined in the methods- i.e. it is discussed that it will be measured using accelerometery, but it is not clear what exact metric you are considering your primary outcome i.e. daily steps vs. daily active energy expenditure.

Lastly, the qualitative analysis lacks sufficient detail in reporting which decreases confidence in the rigour of this aspect of the study.

Strengths:

-well-designed study

-compelling and clear writing

Limitations:

-online exercise is not well reported

-main outcome measure of “habitual physical activity” is not defined in methods.

-some challenges with qualitative analysis reporting

-blinding not possible (acknowledged in limitations)

-small sample size (acknowledged in limitations)

We thank the reviewer for their comments and feedback. We respectfully disagree that habitual physical activity is poorly defined. Physical activity is defined as any bodily movement that uses skeletal muscles and increases energy expenditure. Thus if one were to take more daily steps and increase their daily energey expenditure, by definition, there would be an increase in habitual physical activity.

Line by line feedback:

Line 96: “ability-modified, low-risk exercise program”- this is the only time the exercise program is described in this way. What makes an exercise program high vs. low risk? Typically exercise interventions delivered by physiotherapists are described as “individually tailored” and parameters for how exercises were regressed or progressed are described.

Thank you; we have removed ability-modified, low-risk as a description of our program. We have included more details on the intervention under the Intervention subheading in line with the CERT guidelines. 

Line 205: “All participants (ACTIVE and CON) who completed the online exercise program were invited via email to participate in an optional one-on-one 30-minute qualitative interview, regardless of group allocation.” I think it would be helpful to mention a bit more about the waitlist control group- when were these individuals invited to take part in the intervention? I don’t think it is mentioned prior to this statement.

The reviewer makes a good point

---

## [Decision Letter · Decision Letter 1]

17 Oct 2024

A live online exercise program for older adults improves depression and life-space mobility: A mixed-methods pilot randomized controlled trial

PONE-D-24-24624R1

Dear Dr. Phillips,

We’re pleased to inform you that your manuscript has been judged scientifically suitable for publication and will be formally accepted for publication once it meets all outstanding technical requirements.

Kind regards,

Heather Macdonald, Ph.D

Academic Editor

PLOS ONE

Additional Editor Comments (optional):

Thank you for addressing my queries. 

Reviewers' comments:

Reviewer's Responses to Questions

**Comments to the Author**

1. If the authors have adequately addressed your comments raised in a previous round of review and you feel that this manuscript is now acceptable for publication, you may indicate that here to bypass the “Comments to the Author” section, enter your conflict of interest statement in the “Confidential to Editor” section, and submit your "Accept" recommendation.

Reviewer #1: All comments have been addressed

Reviewer #3: All comments have been addressed

2. Is the manuscript technically sound, and do the data support the conclusions?

Reviewer #1: (No Response)

Reviewer #3: Yes

3. Has the statistical analysis been performed appropriately and rigorously? 

Reviewer #1: (No Response)

Reviewer #3: I Don't Know

4. Have the authors made all data underlying the findings in their manuscript fully available?

Reviewer #1: (No Response)

Reviewer #3: Yes

5. Is the manuscript presented in an intelligible fashion and written in standard English?

Reviewer #1: (No Response)

Reviewer #3: Yes

6. Review Comments to the Author

Reviewer #1: (No Response)

Reviewer #3: (No Response)

7. PLOS authors have the option to publish the peer review history of their article (what does this mean?). If published, this will include your full peer review and any attached files.

Reviewer #1: No

Reviewer #3: No

---

## [Editor Report · Acceptance letter]

28 Oct 2024

PONE-D-24-24624R1 

PLOS ONE

Dear Dr. Phillips, 

I'm pleased to inform you that your manuscript has been deemed suitable for publication in PLOS ONE. Congratulations! Your manuscript is now being handed over to our production team.

Kind regards, 

on behalf of

Dr. Heather Macdonald 

Academic Editor

PLOS ONE